# Empowering Mesenchymal Stem Cells for Ocular Degenerative Disorders

**DOI:** 10.3390/ijms20071784

**Published:** 2019-04-10

**Authors:** Shirley Suet Lee Ding, Suresh Kumar Subbiah, Mohammed Safwan Ali Khan, Aisha Farhana, Pooi Ling Mok

**Affiliations:** 1Department of Biomedical Science, Faculty of Medicine and Health Sciences, Universiti Putra Malaysia, Serdang 43400, Malaysia; suetlee.ding@gmail.com (S.S.L.D.); safwan.aucp@gmail.com (M.S.A.K.); 2Department of Medical Microbiology and Parasitology, Faculty of Medicine and Health Sciences, Universiti Putra Malaysia, Serdang 43400, Malaysia; 3Genetics and Regenerative Medicine Research Centre, Universiti Putra Malaysia, Serdang 43400, Malaysia; 4Institute of Bioscience, Universiti Putra Malaysia, Serdang 43400, Malaysia; 5Department of Pharmaceutical Sciences, Irma Lerma Rangel College of Pharmacy, Texas A&M Health Science Center, Texas University, College Station, TX 77843, USA; 6Department of Clinical Laboratory Sciences, College of Applied Medical Sciences, Jouf University, P.O. Box 2014, Sakaka 72442, Aljouf Province, Saudi Arabia; aishafarhana512@gmail.com

**Keywords:** mesenchymal stem cells, ocular disorders, degeneration

## Abstract

Multipotent mesenchymal stem cells (MSCs) have been employed in numerous pre-clinical and clinical settings for various diseases. MSCs have been used in treating degenerative disorders pertaining to the eye, for example, age-related macular degeneration, glaucoma, retinitis pigmentosa, diabetic retinopathy, and optic neuritis. Despite the known therapeutic role and mechanisms of MSCs, low cell precision towards the targeted area and cell survivability at tissue needing repair often resulted in a disparity in therapeutic outcomes. In this review, we will discuss the current and feasible strategy options to enhance treatment outcomes with MSC therapy. We will review the application of various types of biomaterials and advances in nanotechnology, which have been employed on MSCs to augment cellular function and differentiation for improving treatment of visual functions. In addition, several modes of gene delivery into MSCs and the types of associated therapeutic genes that are important for modulation of ocular tissue function and repair will be highlighted.

## 1. Introduction

The human retina is organized into layers of cells comprising of six unique neurons, namely, rod and cone photoreceptors, ganglion cells, bipolar cells, amacrine cells, and horizontal cells [1]. Together, the retinal neurons process visual signals and form a relay of synaptic transmission, known as photo-transduction, to the visual cortex in the brain. The proper working coordination and condition of these neurons are maintained by the retinal pigmented epithelial (RPE) cells. Any injury or pathology in the eye may lead to the death of retinal neurons, mainly photoreceptors and RPE cells. The loss of these cells is non-replaceable and could contribute to irreversible visual impairment or blindness [1]. Hence, most of the current studies have targeted the regeneration of the photoreceptors or engineering functional RPE layers.

The idea of using stem or precursor cells has emerged in the last decade as a leading approach in regenerative medicine to address ocular disease [2,3]. In this context, mesenchymal stem cells (MSCs) are the most favored candidates for cellular therapy in the correction of ocular disorders [4], including those diseases that are complicated by fibrosis [5,6]. MSCs is a type of adult stem cell which is capable of renewing itself and differentiating into multiple functional cell phenotypes, such as bone, cartilage, fat cells, and others [7]. MSCs were initially discovered in the bone marrow, however, further studies have reported successful isolation and cell expansion from other sources, such as umbilical cord Wharton’s jelly [8,9,10], amniotic fluid [11], dental pulp [12], and adipose tissue [13]. MSC from these origins circumvent the invasive isolation procedure of conventional bone marrow MSCs and are considered hypoimmunogenic [1], thus supporting the use of allogeneic MSCs in regenerative medicine.

Although MSCs are characterized by the expression of a classical set of cell surface antigens (CD90, CD73, CD105, and CD44) [1] and display multilineage differentiation potential, studies reported that different cell sources exhibit unique biological and molecular identities [14,15]. In a recent study, flow cytometric analysis displayed a variation in CD106 (VCAM-1; immunomodulatory effect) expression in different sources of MSC. For example, the expression of CD106 was found to be present in 81% of the MSCs population isolated from the chorionic plate, while dental pulp MSCs showed an absence of this marker [15]. In addition, the author also noticed a discrepancy in the secretion of cytokines. The hepatocyte growth factor (HGF) and transforming growth factor-beta 1 (TGF-β1) were highly expressed by MSCs derived from the chorionic plate. Meanwhile, angiopoietin-1 (Ang-1) and vascular endothelial growth factor (VEGF) were secreted largely by dental pulp MSCs [15]. Thus, it is crucial to understand these biological disparities before selecting the best source for cell isolation in order to tackle different pathologies in the eye.

The possible cellular mechanisms utilized by MSCs in correcting ocular disorders have been intensively reviewed. We also described that MSCs could either directly differentiate into retinal neuron cells or stimulate tissue repair by protecting them from further cell apoptosis, modulation of inflammation, and angiogenesis through its secretory molecules [1]. For example, a study by Sun et al. reported that MSCs grafted in retinal degeneration 1 (rd1) mice could intervene photoreceptor cell apoptosis under the influence of MSCs secretion of pigment epithelium-derived factor (PEDF) [16]. In a rat model of ocular hypertension, the administration of MSCs was reported to relieve intraocular pressure and enhance progenitor cell proliferation [17]. Furthermore, a study demonstrated the generation of photoreceptor-like cells through the direct culture of MSCs with the conditioning medium derived from RPE cultures [18]. Using an MSC/RPE co-culture system, Duan et al. (2013) also evidenced that MSCs were able to adopt the physical and functional characteristics of RPE cells, as observed by the significant expression of CRALBP, RPE65, and ZO-1, and the phagocytosis of photoreceptor outer segments [19].

In our recent review [20], we highlighted the limitations of the current management of eye infection using anti-inflammatory and antimicrobial drugs and surgical approaches. In the same review, we discussed that MSCs excrete human cathelicidin antimicrobial peptide-18 (hCAP18), which has been clinically tested for the treatment of infectious meningitis. It was reported that this peptide molecule could provide protection against infection by viruses, fungus, Gram-negative (*Escherichia coli* and *Pseudomonas aeruginosa*), and Gram-positive (*Staphylococcus aureus*) bacteria. Moreover, MSCs are currently being evaluated for the treatment of organ dysfunction associated with sepsis, including cytomegalovirus infection in clinical settings [20].

Notwithstanding the therapeutic potentials of MSCs, several issues have been raised about current conventional approaches (Figure 1), whereby cells administered in an aqueous medium generally resulted in poor transplanted cell survivability [21,22]. Direct MSC transplantation also yielded unspecific dispersion of cells at the site of injection [23] that could indirectly hamper MSC therapeutic outcomes. The method used for the culture expansion of MSCs prior to administration could also impact the treatment efficiency. For example, a hypoxic culture condition was shown to produce a smaller cell size [24] with improved migration compared to a normoxic culture condition [24]. A substantial advance in our understanding of the regulatory machinery and beneficial secretory proteins of MSCs have paved the way for further development of the technique. Harnessing the potential of biomaterials and tissue engineering [12,25,26,27,28,29], nanotechnology [30,31,32,33], and genome engineering [10,34,35,36,37,38,39,40,41,42,43] to maximize MSCs therapeutic insight for stem cell replacement therapy holds potential for further leaps in using MSC in stem cell therapy. For a clinical translatable stem cell therapy for ocular degenerative disorders, integration of tissue engineering approaches will overcome limitations associated with low transplanted cell survivability [21,22] and cell dispersion [23], and further encourage a targeted delivery system in the transplanted MSCs.

## 2. The Chemistry of Biomaterials and Tissue Engineering in MSC Replacement Therapy

The incorporation of bioengineered scaffolds (also referred as matrices or constructs) in stem cell therapy has emerged as an artificial supporting platform that emulates the physiological niche of the transplanted cells and the biological response of the recipient [44]. Given that the composition that makes up the scaffold bypasses enzymatic degradation in the human body, it can be selectively tailored to mimic the endogenous extracellular matrix by providing a feasible delivery system for MSCs and other essential biomolecules [26,44,45]. These scaffolds are derived from biomaterials of either natural (collagen, fibrin, silk, hyaluronic acid) [12,25,46,47,48] or synthetic (poly (D, L-lactic-co-glycolic acid); PLGA, poly(methyl methacrylate); PMMA, poly(ε-caprolactone); PCL) [49,50,51] origin, which forms a three-dimensional (3D) structure made of the interconnected network.

Several substrates incorporated into condition MSCs from procurement to transplantation [52,53] were shown to strengthen cell-to-cell and cell-to-biomaterial interactions [44,54] and further guided MSC differentiation through the action of local chemical cues [55] (Figure 2). It has been suggested that the topography, mechanical stresses, biocompatibility, degradability, and elasticity of nanomaterials on which the cells adhere will greatly affect cellular functions and differentiation potentials [26,44,45]. A recent study reported that MSCs embedded on a synthetic nanofiber scaffold resulted in the attenuation of oxidative damage in the model of alkali-induced degenerating rabbit corneal epithelium [25]. Following to MSC-nanofiber scaffold transplantation, the author observed a profound reduction in the activity of pro-inflammatory cytokines, including matrix metallopeptidase 9 (MMP-9), inducible nitric oxide synthase (iNOS), and vascular endothelial growth factor (VEGF) along with the decline in corneal transparency and thickness as compared to MSCs transplanted alone [25].

With regard to naturally-derived polymer, intravitreal delivery of MSCs encapsulated in a biodegradable hyaluronic acid-based hydrogel was found to attenuate vascular injury and rescue retinal ganglion cell (RGC) from cell death in the model of retinal ischemia-reperfusion [26]. It was reported that the suppressive effect of MSC was mediated by the downregulation of pro-inflammatory cytokines activity [26]. While MSCs delivered in PBS were mainly dispersed in the vitreous body, the transplanted MSCs embedded in the hydrogel scaffold predominantly localized around the basal membrane of the Műller glia and concomitantly induced the local release of neurotrophic factors, including nerve growth factor (NGF) and brain-derived neurotrophic factor (BDNF) from Műller glia cells [26].

Meanwhile, several studies evinced that probing MSCs with polymeric scaffolds improved the regeneration capacity of MSCs into the desired retinal cell types. For example, bio-compatible fibrin hydrogel was observed to direct dental pulp-derived MSCs cell fate into retinal ganglion-like cells in vitro [12]. The differentiated MSCs cultured in induction medium supplemented with fibroblast growth factor 2 (FGF2), sonic hedgehog (Shh), and fetal bovine serum (FBS) were observed to have an increased expression of transcription factors essential for RGC cell fate specification, such as paired box protein 6 (Pax6), atonal bHLH transcription factor 7 (Atoh7), and brain-specific transcription factor 3b (Brn3b), as compared to their culture in the absence of hydrogel [12]. Similar finding previously illustrated an enhanced differentiation potential of bone marrow-derived MSCs into retinal-like neurons following culture induced with a biopolymer-based scaffold made up of silk fibroin-conjugated with integrin-binding laminin peptide motifs under retinoic acid stimulation [27]. This approach was found to circumvent shortfalls of using whole laminin protein, such as poor stability and cost, while it exemplified the MSC adherence and proliferation rate in the presence of glycine amino acid found within silk fibroin [27]. More recently, it was suggested that MSCs isolated from the trabecular meshwork displayed a greater shift into photoreceptor cell fate when cells were seeded onto the amniotic membrane scaffold, in comparison to conventional polystyrene culture [56]. Thus, with the incorporation of polymeric scaffolds, we have evidenced its use in probing MSC differentiation and reparative effects, which enables an effective transplantation strategy in the future, especially when MSCs were to be delivered into a hypoxic microenvironment [28].

Until recently, the introduction of three-dimensional (3D) bio-printing has been used substantially in various fields of architecture, art, and even in medicine to produce 3D models for medical imaging [57]. Bio-printing employs biological components and living cells in ‘bioinks’ to construct viable 3D structures that closely resemble the anatomy and physiology of the human tissue [58]. Few studies have also explored the use of 3D bio-printing on neural and retinal tissues [59,60,61,62,63]. A study done by Lorber et al. [61] showed that adult rat retinal ganglion cells and glial cells could be 3D-printed using the piezo inkjet printing technology without significant loss in cell viability. Interestingly, Kolesky et al. [64] successfully constructed a heterogeneous population containing cells, extracellular matrix, and even vascular tissue through 3D bio-printing. This method could be adapted to construct vascularized retinal tissues that could better recapitulate the in vivo physiology of the retina. Since the conventional monolayer MSC culture techniques lack the capacity to produce a high number of functional RPE and retinal cells [65], 3D bio-printing can be a useful technique to generate transplantable MSC-derived retinal tissue for the treatment of ocular disorders. Using 3D bio-printing technology, MSCs can be triggered to differentiate into retinal cells and culture on a biomaterial platform for fabrication into a functional 3D retinal tissue structure.

Furthermore, co-printing using a thermal-based approach may serve as an efficient transfection tool to deliver therapeutic agents which may influence cell survivability, proliferative, and regenerative capabilities [66,67,68]. Cui et al. [66] previously observed that cell printing causes a transient pore opening on the cell membrane of printed cells, which was found to facilitate transfection of plasmid encoding for green fluorescent protein (GFP) without compromising cell viability [67,68]. This strategy can be employed on MSCs to achieve a successful delivery system of functional genes or nanocarriers that minimizes issues related to cell incompetency and toxicity [69,70]. The combination of MSCs therapy with bio-printing technology will thus create a patient-specific therapy through the customization of fabricated retinal tissue prototype in the near future and, further supporting the development of targeted therapy in MSCs.

## 3. Crosslinking Nanotechnology with Mesenchymal Stem Cells

Nanotechnology revolutionized the use of technology in physics, chemistry, and biology for the creation of nanoscale materials. The involvement of nanotechnology for cell imaging and therapy has aided researchers to monitor the fate of transplanted cells, as well as enabled the local delivery of growth factors and drugs [30,31]. This could be manipulated to assist in identifying the causes responsible for the discrepancy in therapeutic outcomes between patients when subjected to MSC therapy [71], and hence provide a qualitative and quantitative evaluation of the efficiency of transplantations.

The application of nanotechnology has been demonstrated in pre-clinical settings as a targeted cancer therapy. It is helpful to oncologists tracking the residential and metastasized malignant cells to elucidate treatment efficiency [41,42]. A previous study has shown that gold nanoparticles could selectively target tumor-associated antigen on the cancerous cells through conjugation with specific antibodies. Exposure to photo-thermal energy on the nanoparticles could raise the temperature of the cancerous cells, and hence, it causes the destruction of these heat-sensitive cells. This strategy could prevent unnecessary damage to adjacent healthy tissue, a complication which is usually associated with other treatments such as chemo- or radio-therapy [9,39].

It is noteworthy that the co-labeling of MSCs with nanoparticles may offer a novel strategy for the treatment of eye cancers associated to choroidal melanoma [43] and retinoblastoma [44,45]. MSCs possess the ability to home towards tumor cells [46,47,48], and the destruction of MSCs can lead to the release of beneficial cytokines and trophic factors [49,50,51], which would further promote the local recovery and regeneration of injured tissue. A recent study has examined the feasibility of labeling MSCs with gold nanoparticles prior to subretinal transplantation into a rat model [9]. The authors indicated that there was no physical alteration in labeled MSCs which allowed real-time monitoring of the cell localization using micro-computed tomography [9]. The combinational therapy using nanoparticles and MSCs may thus provide a fundamental approach to achieve a synergistic effect for the treatment of ocular cancer.

Alternatively, magnetic-based nanoparticles have been considered as a potent cell or drug carrier as their bioreactive surfaces can be formulated to attain a stronger interaction towards the targeted site while imposing minimal damage to the healthy tissue [72]. Ferumoxytol is a type of SPIO nanoparticle which has long been approved by the US Federal Drug Administration for use in anemia patients [73]. Previous studes have demonstrated successful tracking of engrafted MSCs in the rat model of optic nerve crush by the use of superparamagnetic iron oxide (SPIO) nanoparticles and magnetic resonance imaging [11]. It was detected that MSCs mainly resided around the injured sites of the vitreous body and optic nerve, and thus permits long-term assessment of MSCs in vivo [11]. According to Liu et al., ferumoxytol can be used to establish a new strategy for labeling MSCs and that cell labeling is dependent on MSC cell size [71]. Nevertheless, it was shown that MSC pre-labeled with ferumoxytol displayed a relatively high engulfment by macrophages upon in vivo administration into a rat model of cartilage defect with greater phagocytosis in apoptotic MSCs, which releases iron oxide. These intracellular nanoparticle delivery systems are typically not high-throughput [74] and have been shown to cause significant cellular injury and death [75].

## 4. Genetic Modifications to Deliver Therapeutic Genes

Incorporation of gene editing technology into stem cells for the treatment of ocular disorder due to defective genes or to correct dysregulation of gene expression has seen several successes [36,37]. Delivery of therapeutic gene into MSCs requires either viral [32,33,75] or non-viral [40] transfection methods. Here, we reviewed some of the strategies and choice of therapeutic genes which had been used to restore the RGCs, photoreceptors, or RPE cells.

Ample studies have attempted to deliver neuroprotective genes, such as *BDNF* and *PDGF*, into MSCs through viral transduction. For instances, Harper et al. [35,43] reported that co-treatment of BDNF-transduced MSCs with glutamate- and hydrogen peroxide-induced RGCs were found to prevent RGC from cell death and further promote neurite growth in cultured RGCs. The presence of BDNF receptor, tropomyosin receptor kinase B (TrkB), on the RGCs have previously been found to mediate RGCs neuroprotection. The research showed that the transplantation of BDNF-transduced MSCs was able to survive and secrete functional BDNF protein for the enhancement of RGC viability in a chronic glaucomatous rat model [35]. It was also demonstrated that MSCs transduced with Math5 (Atoh) adenoviral vector were found to facilitate MSC differentiation into RGC-like cells, characterized by the expression of RGC-related genes, such as *GAP-43* and *Brn3b* [76].

Other than the restoration of RGCs, lentiviral-transduced MSCs could also migrate, integrate, and stably express pro-survival neurotrophin-4 (NT-4) at the injured retina, particularly on the RPE and photoreceptor cells [41]. Moreover, transplanted MSCs encoding NT-4 were found to restore retinal function and architecture through upregulation of anti-apoptotic mediators including B cell lymphoma-2 (Bcl-2) and baculovirus inhibitor-of-apoptosis repeat containing (BIRC) proteins via activation of mitogen-activated protein kinase (MAPK) and Akt signaling cascades, and the induction of crystallins for neurogenesis [41].

While there is debate on the risk of tumorigenicity as a result of viral gene integration into proto-oncogene site in cells, Boura et al. [42] have noticed that modification of MSCs using lentiviral-based delivery of HLA-G was found to enrich MSC immunomodulatory actions which are absent in non-viral transfer. It was also shown that the lentiviral approach significantly prevented the activation of lethal immune responses toward MSCs through the re-establishment of immune tolerance against NK cells and T cells proliferative responses [42]. Hence, this would further promote the sustainability of transplanted MSCs for tissue repair.

Of note, studies showed that pre-conditioning of the microenvironment with growth factors such as erythropoietin (EPO) before stem cell transplantation could improve cell survival [77,78,79,80] and tissue repair with a smaller dose of transplanted cells. The manipulation of MSCs to deliver EPO for the treatment of ocular disorders by direct injection into the vein is feasible in the future as these cells could migrate to the inflammatory site [81,82] and cross the blood-retinal barrier (BRB) [83,84,85,86,87,88]. The autocrine activity of EPO itself on the stem cells could enhance the survivability of transplanted cells [89,90] in a pathologically-harsh microenvironment. Compared to unmodified MSCs transplantation, Guan et al. [91] discovered a significant improvement on the retinal morphology and function following subretinal transplantation of *EPO* gene-modified MSCs in a rat model of retinal degeneration (RD) [91]. Despite that, there is a need to research for a better-controlled regulation system on the expression of the EPO gene in MSCs for ocular therapy in the future [92,93]. It is noteworthy that the utilization of a short DNA construct vector known as minimalistic, immunologically defined gene expression (MIDGE) has been shown to be relatively safer, yet capable of providing stable and prolonged EPO protein secretion when transfected into human bone marrow MSCs in vitro [40]. Other methods such as ultrasound- [94] or microbubbles-assisted [95] gene delivery could be used to improve transfection efficiency [88,89,90] and promote MSCs trans-migratory capability [96,97,98], and its differentiation potential too [99].

## 5. Conclusions

Accumulative pre-clinical and clinical trials have reported beneficial outcomes using MSCs for a wide range of pathological complications pertaining to ocular degenerative disorders. Hence, it has been considered as a source for cell replacement therapy. We have reviewed several recent approaches to maximizing the native therapeutic potential of MSCs, especially to overcome complications concerning low transplanted cell viability and unspecific cell targeting to the damaged site. Reports of disconcerting outcomes have warranted more provision of a standardized method to assess the kinetic rate of the biomaterial degradation, the toxicity level of administered nanoparticles, as well as the mode of gene delivery that minimize the chances for tumor formation in MSCs in the future.

## Figures and Tables

**Figure 1 ijms-20-01784-f001:**
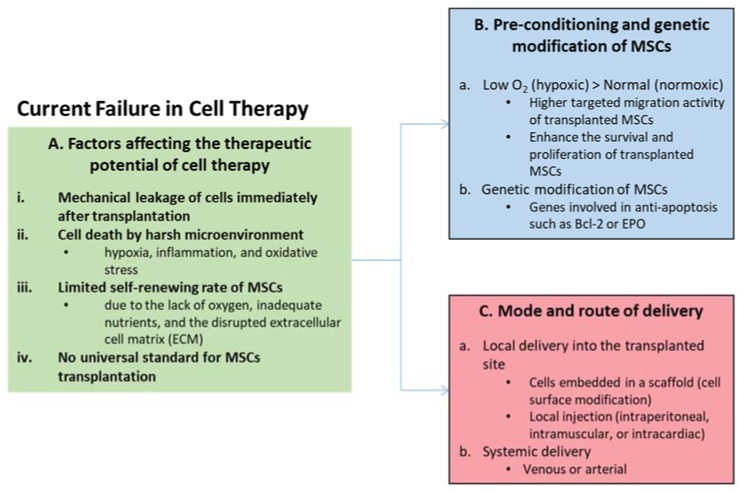
Current failures in cell therapy. (**A**) The patient’s age and mechanical and pathological conditions are among the factors that affect the therapeutic potential of cell therapy. In particular, transplanted cell survival may be affected by (i) mechanical stress during the transplantation procedure; (ii) a harsh microenvironment due to the activation of inflammation-related factors; (iii) oxygen and nutrient starvation due to poorly vascularized environments at the site of implantation; and (iv) a lack of optimization of the delivery protocols. (**B**) The benefits of cell transplantation could be improved by donor cell preconditioning or modifying transplanted cells prior to implantation to support or enhance their resistance to hypoxic stress. (**C**) The tissue engineering approach could enhance the survival of transplanted cells through the use of suitable biomaterials as carriers, such as a biologic-derived ECM scaffold. O_2_ = oxygen; Bcl-2 = B-cell lymphoma 2; EPO = erythropoietin.

**Figure 2 ijms-20-01784-f002:**
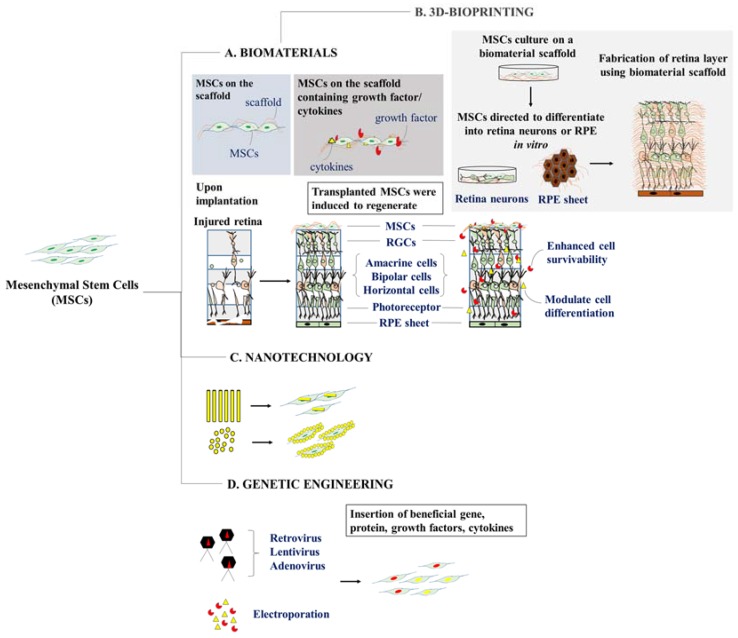
Strategies for empowering mesenchymal stem cells for ocular degenerative disorders. (**A**) The development of biomaterials can be utilized with or without the addition of growth factors or cytokines that may selectively promote multipotent mesenchymal stem cells (MSCs) either to restore or differentiate into desired cells. (**B**) Biomaterial can also be 3D-printed to form a sheet or layer of cells that resembles the local environment of the damaged site (Right). (**C**) MSCs can also be encapsulated or coated with nanoparticles of various sizes or origins to enhance the native property of MSCs. (**D**) Genetic modification of MSCs can be achieved by introducing MSCs with genes containing beneficial trophic factors or cytokines that could affect the physiological behavior of MSCs.

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
