# Peer review of "Empowering Mesenchymal Stem Cells for Ocular Degenerative Disorders"

_ijms, 2019, doi:10.3390/ijms20071784_

Round 1

Reviewer 1 Report

Dear authors

The argument regarding the importance of MSCs for the possible treatment of ocular degenerative disorders focusing on various types of biomaterials and advances in nanotechnology that facilitate and improve the application of MSCs in such diseases is of real interest.

In the complex, the manuscript give an interesting contribute about the role of MSCs in the therapy of ocular degenerative disorders albeit some flaws are present in the manuscript. In particular, the manuscript should be revised in the English grammar because several mistakes in verb form (singular and plural) are present in different sentences distributed in the manuscript. In addition, in the introduction ref. 6,7 should be changed with references more adequate to the context. In addition, the authors should be add more reference recently published on the theme (Stern ET AL., Regenerating Eye Tissues  to Preserve and Restore Vision,  Cell Stem Cell 22, June 1, 2018)

Author Response

We agree to Reviewer 1 and have amended the manuscript according to your suggestion.

We have revised the English grammar for the whole manuscript and replaced ref. 6,7 (Fellows CR, Matta C, Zakany R, Khan IM, Mobasheri A. Adipose, bone marrow and synovial joint-derived mesenchymal stem cells for cartilage repair [Internet]. Vol. 7, Frontiers in Genetics. Frontiers Media SA; 2016 [cited 2019 Mar 11]. p. 213.).

Reviewer 2 Report

The review covered the topic quite nicely and was accurate and concise.

It would be beneficial to include a paragraph in the beginning describing the different cell types of the eye and to highlight the main cell types being targeted for treatment.

Is there evidence that the retinal neurons formed or the retinal like cells are actually functional? if so how is this assessed?

It would also add to the review if you could describe the MSC populations in abit more detail and which ones are better for therapy. For example what cell surface markers do they have and do distinct populations with specific cell surface markers have a greater effect in terms of therapy?

Author Response

Response to Reviewer 2 Comments

We thank Reviewer 2 for the comments and suggestions.

Point 1: The review covered the topic quite nicely and was accurate and concise. It would be beneficial to include a paragraph in the beginning describing the different cell types of the eye and to highlight the main cell types being targeted for treatment.

Response 1: Page 3, line 74 – 82. The human retina is organized in layers of cells comprising of six unique neurons namely, rod and cone photoreceptors, ganglion cells, bipolar cells, amacrine cells, and horizontal cells (1).  Together, the retinal neurons process visual signals and form a relay of synaptic transmission, known as photo-transduction, to the visual cortex in the brain. The proper working coordination and condition of these neurons are maintained by the retinal pigmented epithelial (RPE) cells. Any injury or pathology in the eye may lead to the death of retinal neurons, mainly photoreceptors, and RPE cells. The loss of these cells are non-replaceable, and could attribute to irreversible visual impairment or blindness (1). Hence, most of the current studies have targeted on the regeneration of the photoreceptors or engineering functional RPE layers.

Point 2: Is there evidence that the retinal neurons formed or the retinal like cells are actually functional? if so how is this assessed?

Response 2: Page 4, line 124 – 126

Using a MSC/RPE co-culture system, Duan et al. (2013) also evidenced that MSCs were able to adopt the physical and functional characteristics of RPE cells, as observed by the significant expression of CRALBP, RPE65 and ZO-1, and phagocytosis of photoreceptor outer segments (19).

Point 3: It would also add to the review if you could describe the MSC populations in abit more detail and which ones are better for therapy. For example what cell surface markers do they have and do distinct populations with specific cell surface markers have a greater effect in terms of therapy?

Response 3: Page 3, line 89 – 112

MSCs is a type of adult stem cells which is capable of renewing themselves and differentiating into multiple functional cell phenotypes, such as bone, cartilage, fat cells, and others (7). MSCs were initially discovered in the bone marrow, however, further studies have reported successful isolation and cell expansion from other sources too such as umbilical cord Wharton’s jelly (8–10), amniotic fluid (11), dental pulp (12) and adipose tissue (13). MSC from these origins circumvent the invasive isolation procedure of conventional bone marrow MSCs and are considered hypoimmunogenic (1), thus supporting the use of allogeneic MSCs in regenerative medicine.

Although MSCs are characterized by the expression of classical set of cell surface antigens (CD90, CD73, CD105, and CD44) (1) and displaying multilineage differentiation potential, studies reported that different cell sources exhibit unique biological and molecular identities (14,15). In a recent study, flow cytometric analysis displayed a variation in CD106 (VCAM-1; immunomodulatory effect) expression in different sources of MSC. For example, the expression of CD106 was found to be present in 81% of the MSCs population isolated from the chorionic plate, while dental pulp MSCs showed absence of this marker (15). In addition, the author also noticed a discrepancy in the secretion of cytokines. The hepatocyte growth factor (HGF) and transforming growth factor-beta 1 (TGF-β1) were highly expressed by the MSCs derived from the chorionic plate. Meanwhile, the angiopoietin-1 (Ang-1) and vascular endothelial growth factor (VEGF) were secreted largely by the dental pulp MSCs (15). Thus, it is crucial to understand these biological disparities before selecting the best source for cell isolation in order to tackle different pathologies in the eye.